# Towards out-of-distribution generalizable predictions of chemical kinetic properties

**Zihao Wang**[*]
CSE, HKUST
zwanggc@cse.ust.hk

**Yongqiang Chen**[*]
CSE, CUHK
yqchen@cse.cuhk.edu.hk

**Yang Duan, Weijiang Li**
CS, UIUC
{yangd4,wl13}@illinois.edu

**Bo Han**
CS, HKBU
bhanml@comp.hkbu.edu.hk

**James Cheng**
CSE, CUHK
jcheng@cse.cuhk.edu.hk

**Hanghang Tong**
CS, UIUC
htong@illinois.edu

## Abstract

Machine Learning (ML) techniques have found applications in estimating chemical kinetic properties. With the accumulated drug molecules identified through "AI4drug discovery", the next imperative lies in AI-driven design for high-throughput chemical synthesis processes, with the estimation of properties of unseen reactions with unexplored molecules. To this end, the existing ML approaches for kinetics property prediction are required to be Out-Of-Distribution (OOD) generalizable. In this paper, we categorize the OOD kinetic property prediction into three levels (structure, condition, and mechanism), revealing unique aspects of such problems. Under this framework, we create comprehensive datasets to benchmark (1) the state-of-the-art ML approaches for reaction prediction in the OOD setting and (2) the state-of-the-art graph OOD methods in kinetics property prediction problems. Our results demonstrated the challenges and opportunities in OOD kinetics property prediction. Our datasets and benchmarks can further support research in this direction. The github repository for code and data can be found in https://github.com/zihao-wang/ReactionOOD.

## 1 Introduction

In recent years, graph machine learning has been widely used in scientific discovery [Wang et al., 2023, Zhang et al., 2023] and gained particular success in chemistry [Gilmer et al., 2017, Jumper et al., 2021, Mullowney et al., 2023]. The underlying rationale is the long-standing structure-property relationship [Mihalić and Trinajstić, 1992] in chemistry. For example, Graph Neural Networks (GNNs) can efficiently encode information at both the molecular structure level and the atom level within a molecule, which reveal compelling properties of the molecule [Gilmer et al., 2017]. Such methods yield efficient, cheap, but still effective predictions of the properties of *unseen molecules* before expensive experiments or computations, which can serve as valuable reference information for drug discovery [Mullowney et al., 2023].

One of the *next* questions to be answered after the discovery of a proper but unseen molecule is *How to efficiently obtain unseen molecules through chemical synthesis.* In contrast to the molecule property estimation problem that concerns a *single* molecule, chemical synthesis processes involve the proper arrangement of various reactions that encompass *multiple* molecules under optimal conditions. Therefore, the very first step of achieving a high-throughput synthesis of unseen molecules is to estimate the properties of chemical reactions [Warr, 2014], especially kinetics properties that

---

[*]Equal contribution.

NeurIPS 2023 Workshop in AI for Scientific Discovery: From Theory to Practice.

describe the "rate" of reactions. This prediction task is expected to be Out-Of-Distribution (OOD) generalizable so that the kinetic properties of OOD reactions with unseen molecules can be well predicted.

Recently, machine learning methods have been applied to predict the kinetic properties of reactions [Heid and Green, 2021]. In existing studies, chemical reactions are assumed to be Independently and Identically Distributed (IID), and models are trained and tested within random splits [Heid and Green, 2021, Stuyver and Coley, 2022, Heid et al., 2023]. However, results from such IID assumptions provide little credible insight into the performances of existing ML methods in OOD reaction property prediction. Meanwhile, existing theoretical and empirical studies for OOD generalization on graphs [Ji et al., 2022, Gui et al., 2022], are restricted to problems with a single graph. How OOD methods perform reaction properties prediction with multiple molecules is still unknown.

To fill this gap, this paper discusses the out-of-distribution generalization issue when applying machine learning methods to the prediction of chemical reaction properties. We propose three levels of OOD shifts for ML-based reaction prediction: Structure OOD, Mechanism OOD, and Conditional OOD. Then, we reorganize recent reaction kinetic databases [Johnson et al., 2022] and create a comprehensive dataset in three levels of OOD. Furthermore, we empirically justify the performance of state-of-the-art kinetic property prediction produced by state-of-the-art OOD methods for general and graph inputs. Our results demonstrated that there remain huge ID-OOD performance gaps under different distribution shifts in chemical reactions for existing OOD methods.

## 2    Related works

Increasing efforts have been made to devise machine learning approaches for various aspects of chemical reaction systems [Davies, 2019, Stocker et al., 2020, Meuwly, 2021, Strieth-Kalthoff et al., 2022], such as reaction classification [Schwaller et al., 2021b, Burés and Larrosa, 2023], reaction optimization [Felton et al., 2021], atom mapping [Schwaller et al., 2021a], and the most fundamentally, reaction property prediction [Heid and Green, 2021]. With the burst of chemical reaction data [von Rudorff et al., 2020, Spiekermann et al., 2022, Johnson et al., 2022, Choi, 2023, Stuyver et al., 2023, Zhao et al., 2023], the Graph Neural Network (GNN) based methods [Heid and Green, 2021, Stuyver and Coley, 2022, Heid et al., 2023] are demonstrated its clear advantage over traditional methods by leveraging the structure of reactants and products.

Out-of-distribution shift is one of the long-standing problems in machine learning [Vapnik, 1991, Quinonero-Candela et al., 2008, Shen et al., 2021]. Recently, OOD generalizable graph neural networks have been discussed extensively [Bevilacqua et al., 2021, Zhu et al., 2021, Wu et al., 2022b,a, Chen et al., 2022]. When it comes to scientific discovery, out-of-distribution generalization capabilities enable machine learning methods to find more reliable discoveries from existing observations. A thorough investigation of the intersection of OOD and drug discovery can be found at Ji et al. [2022] and Gui et al. [2022]. However, as we will identify in the incoming parts, the out-of-distribution shifts for chemical reactions are radically different from those with existing graph OOD settings [Gui et al., 2022], and existing OOD methods do not perform well.

## 3    Preliminary

### 3.1    Chemical reactions and kinetic property prediction

A chemical reaction $\mathfrak{R}$ is described by the reactants $r_1, \ldots, r_m$, products $p_1, \ldots, p_n$, and the conditions $c_1, \ldots, c_l$.[2]

$$r_1 + \cdots + r_m \overset{c_1,\ldots,c_l}{\Longrightarrow} p_1 + \cdots + p_n. \tag{1}$$

Multiple molecules (including atoms, ions, and other species) are involved in one reaction, which differs from molecule property prediction tasks where only one molecule structure is considered.[3]

---

[2]In chemistry literature, different arrow types indicate different reaction types. We use $\Longrightarrow$ for simplicity.

[3]In this paper, we assume reactions adhere to the law of conservation of matter, ensuring both atoms and electrons are equally represented on either side of the reaction equation.

Table 1: Two types of distribution shifts. $[X]$ denotes the domain index of the input $X$.

| Distribution shifts | $P([X])$ | $P(Y|[X])$ |
|---|---|---|
| Covariate shift | $P^{\text{train}}([X]) \neq P^{\text{test}}([X])$ | $P^{\text{train}}(Y|[X]) = P^{\text{test}}(Y|[X])$ |
| Concept shift | $P^{\text{train}}([X]) = P^{\text{test}}([X])$ | $P^{\text{train}}(Y|[X]) \neq P^{\text{test}}(Y|[X])$ |

Each molecule $r_i$, $p_j$ is considered as a molecule graph, with atoms as attributed nodes and bonds as attributed edges. For a chemical reaction $\mathfrak{R}$, $R[\mathfrak{R}]$ ($P[\mathfrak{R}]$) denotes the graph of reactants (products) as the union of the reactant (product) molecule graphs. $C[\mathfrak{R}]$ denotes the set of conditions. $\mathcal{G}_R$ ($\mathcal{G}_P$) and $\mathcal{C}$ denote the space of graphs for reactants (products) and the space of conditions, respectively.

Kinetics properties investigated in existing literature include *reaction barrier* (activation energy) [von Rudorff et al., 2020, Spiekermann et al., 2022, Stuyver et al., 2023, Zhao et al., 2023] and *rate constants* [Heid and Green, 2021]. As one of the simplest mathematical descriptions of the rate constant, the Arrhenius equation reads

$$k = A \exp\left(\frac{-E_a}{RT}\right), \tag{2}$$

where $k$ is the rate constant, which is modeled by a function of the Arrhenius factor $A$, the reaction barrier (activation energy) $E_a$, the universal gas constant $R$, and the absolute temperate $T$. Though Equation (2) only holds under mild conditions for some reactions, we could still summarize that, the rate constant $k$ is jointly described from three aspects: (1) the inherent property of the reaction, such as the barrier $E_a$ [Donahue, 2003], (2) the conditions, such as the temperature $T$, and (3) other parameters $A$ to fit the empirical data.

In this paper, we also consider the reaction barrier $E_a$ and the rate constant $k$ as the target of prediction. Let $\mathcal{Y}$ be the space of the target property. The target $Y$ of reaction $\mathfrak{R}$ is computed by the learnable function $f_\theta : \mathcal{X} \mapsto \mathcal{Y}$, where $\mathcal{X} = \mathcal{G}_R \times \mathcal{G}_P \times \mathcal{C}$ is the input space.

## 3.2 OOD shifts

Out-of-distribution generalization is one of the key topics in machine learning research. It considers the shifts of the joint distribution $P(X, Y)$ over $\mathcal{X} \times \mathcal{Y}$ during training and testing phase, $P^{\text{train}}(X, Y) \neq P^{\text{test}}(X, Y)$ [Gui et al., 2022]. Applying the conditional probability formula,

$$\underbrace{P^{\text{train}}(Y|X)}_{\text{Concept}} \underbrace{P^{\text{train}}(X)}_{\text{Covariate}} \neq \underbrace{P^{\text{test}}(Y|X)}_{\text{Concept}} \underbrace{P^{\text{test}}(X)}_{\text{Covariate}}. \tag{3}$$

This decomposition introduces two basic types of OOD shifts: covariate shift and concept shift, where one part of the distribution is changed while the other part is fixed.

One common way to characterize distribution shifts in high-dimensional input space is to properly separate the spaces into domains. Let $\mathcal{D}$ be the set of domains and $\mathcal{X}$ be the input space. For each input $X$, we define its domain index $[X]$. Table 1 summarizes the covariate shift and concept shift with the domain index.

Throughout this paper, the domain index $[X]$ is equivalent to an equivalent relation $\sim$. Specifically, for $X', X \in \mathcal{X}$, $[X'] = [X]$ if and only if $X' \sim X$. Then, it suffices to define **the equivalent relation** over the input space, which will naturally lead to the definition of the domain index and then two kinds of OOD shifts.

## 4 Three levels of OOD shifts in kinetic property prediction

This section defines three levels of OOD settings and details the OOD domains by constructing the **equivalent relations** in the input space $\mathcal{X} = \mathcal{G}_R \times \mathcal{G}_P \times \mathcal{C}$. The definition of equivalent classes, so as the OOD shifts, can be decomposed into the subspaces of $\mathcal{X}$. Structure OOD (S-OOD) establishes the equivalent relation in the subspace $\mathcal{G}_R \times \mathcal{G}_P$ while Condition OOD (C-OOD) concerns the equivalent relation in the subspace $\mathcal{C}$. Mechanism OOD (M-OOD) considers the shifts in general chemical reaction space $\mathcal{X}$ with the expertise of chemists. Figure 1 indicates the different levels of out-of-distribution generalization.

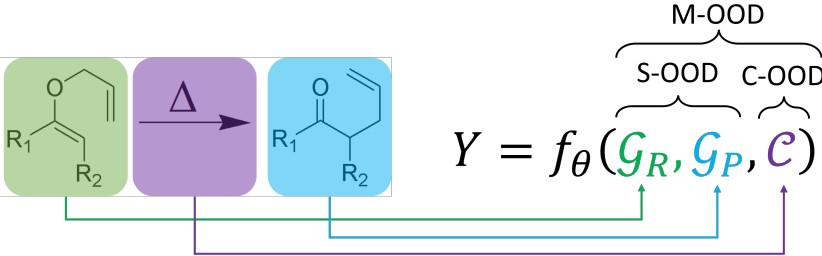

Figure 1: Illustration of different levels of reaction OOD. S-OOD and C-OOD focus on the distribution shifts in the subspace of the input. M-OOD focuses on the distribution shifts in the entire reaction space.

## 4.1 Structure OOD (S-OOD)

The first level of OOD settings is about the graphs for reactants and products, which are directly related to the graph machine learning methods such as graph neural networks.

**Definition 1** (Structure OOD). *Let $\sim$ be an equivalent relation in $\mathcal{G}_R \times \mathcal{G}_P$ and $[X]$ be the equivalent class containing $X$ induced by $\sim$. We say $\sim$ defines a structure OOD setting.*

By instantiating the equivalent relation $\sim$, we can easily define several OOD settings. We begin with the basic cardinality $|G|$ of graph $G$.

**Example 1** (S-OOD by total atom number). *Consider the equivalent relation $\sim$, such that $R_1 \times P_1 \sim R_2 \times P_2$ if and only if $|R_1| = |R_2|$, i.e., the total number of atoms in two reactions are equal. Then the induced S-OOD setting is about the atom number of the largest reactant.*

Besides, the size of each reactant molecule also plays an important role. For example, the number of carbon atoms in an organic compound is related to its phase status in the ambient temperature and pressure, thus affecting the chemical reactions. Inspired by this, we can also define another type of S-OOD.

**Example 2** (S-OOD by the atom number of the largest reactant). *Consider the equivalent relation $\sim$, such that $R_1 \times P_1 \sim R_2 \times P_2$ if and only if $\max_{r_i \in R_1} |r_i| = \max_{r_j \in R_2} |r_j|$, i.e., the total number of atoms in two reactions are equal. Then the induced S-OOD setting is about the total atom number.*

Furthermore, it is also possible to define the S-OOD by considering the scaffold of molecules.

**Example 3** (Structure OOD by the scaffold of the first reactant). *Let $\sim_S$ be the equivalent relation for two molecular graphs with the same scaffold. Consider the equivalent relation $\sim$ such that $R_1 \times P_1 \sim R_2 \times P_2$ if and only if $r_{11} \sim_S r_{21}$ for $r_{11} \in R_1$ and $r_{21} \in R_2$. Then the induced OOD setting is about the first reactant scaffold.*

One can easily induce a more complex S-OOD setting by using the scaffold equivalence.

**Example 4** (Structure OOD by the mutual scaffold of all reactants). *Let $\sim_S$ be the equivalent relation for two molecular graphs with the same scaffold. Consider the equivalent relation $\sim$ such that $R_1 \times P_1 \sim R_2 \times P_2$ if and only if $\exists r_i \in R_1, r_j \in R_2, r_i \sim_S r_j$. Then the induced OOD setting is about the reactant scaffold.*

We could see that basic properties of molecules such as atom number and scaffold can define various types of S-OOD for chemical reactions, which is more complex than the situation for single molecule property predictions [Ji et al., 2022].

## 4.2 Condition OOD (C-OOD)

Focusing on the subspace $\mathcal{C}$ of $\mathcal{X}$, C-OOD investigates the generalization problem of the same reaction with respect to the reaction condition. Typical examples of C-OOD include rate-temperature dependency and rate-temperature-pressure dependency.

It should be stressed that Equation (2) is not always sufficient to describe how the rate constant $k$ changes with temperature. It turns out that the reaction rate cannot be summarized by a uni-

Table 2: Summary of datasets. † indicates the database processed by Heid et al. [2023].

| OOD LEVEL | TARGET | DOMAIN | DATABASE | # RXN | # SMP | REFERENCE |
|---|---|---|---|---|---|---|
| S-OOD | BARRIER | TOTAL ATOM NUMBER (EG. 1) | E2 & S$_N$2† | 3625 | 3625 | VON RUDORFF ET AL. [2020] |
| | | | RDB7† | 23852 | 23852 | SPIEKERMANN ET AL. [2022] |
| | | | CYCLOADDITION† | 5269 | 5269 | STUYVER ET AL. [2023] |
| | | FIRST REACTANT SCAFFOLD (EG. 3) | RDB7† | 23852 | 23852 | SPIEKERMANN ET AL. [2022] |
| | | | CYCLOADDITION† | 5269 | 5269 | STUYVER ET AL. [2023] |
| C-OOD | RATE CONSTANT | $T$ | RMG LIB. T | 29161 | 87340 | JOHNSON ET AL. [2022] |
| | | $(T, P)$ | RMG LIB. TP | 3444 | 113695 | |
| M-OOD | BARRIER | MECHANISM | RMG FAMILY | 12129 | 12129 | |

fied formulation. Chemists employ various forms of formulas to describe the dependency between the rate constant and conditions through data-driven approaches such as Chebyshev polynomial fitting [Heal, 1999] or analytical approaches like the Lindemann mechanism [Lindemann et al., 1922]. Other types of rate constant formulas can be found in Johnson et al. [2022].

Due to the complex nature of the dependency between the rate constant and conditions, C-OOD generalizability is particularly important if one wants to obtain the optimal condition of a reaction from machine learning models. Meanwhile, the equivalent relation $\sim$ for C-OOD is easier to construct since the most common conditions are temperature and pressure, which are all real-valued.

### 4.3 Mechanism OOD (M-OOD)

Another important intuition for characterizing chemical reaction systems, in particular for organic reactions, is the reaction mechanism. Reaction mechanisms are expert knowledge about how reactions happen and are examined in long-standing chemistry research. Typical examples include the E2 and S$_N$2 mechanisms [von Rudorff et al., 2020] that are usually considered as widely existing competing mechanisms.

Similar to the definition of S-OOD, M-OOD is also defined by the equivalent relationship, but in the general chemical reaction space $\mathcal{X}$ with an expert-defined relationship $\sim_M$. [4]

**Definition 2** (Mechanism OOD). *Let $\sim_M$ be the mechanism equivalence relationship in $\mathcal{X}$, such that $\mathfrak{R}_1 \sim_M \mathfrak{R}_2$ if and only if $\mathfrak{R}_1$ and $\mathfrak{R}_2$ follow the same established mechanism. We say $\sim_M$ defines a M-OOD setting.*

Reaction mechanisms also heavily rely on the molecular structure of reactants and products. However, M-OOD is motivated very differently from S-OOD by size equivalence or scaffold equivalence. M-OOD follows the well-established understanding and explanations in chemistry research. In this work, we define the M-OOD dataset by the kinetic database built by Johnson et al. [2022]. Detailed introduction can be found in Section 5.

## 5 Datasets and benchmarks

This section presents the details of constructing the benchmark to evaluate OOD generalizability of machine learning methods. We introduce how to create the datasets in different settings and the methods to benchmark. The derived collections of datasets is termed as ReactionOOD, which will be updated following to the growing of reaction kinetic databases.[5]

### 5.1 Dataset construction

Table 2 summarizes the information about the OOD datasets at three levels, with detailed target, domain, database, and statistics. For each database, we reorganize it into one or more domains to form multiple OOD datasets at different OOD levels. It worth mentioning that databases of E2

---

[4]The reaction of complex molecules can involve multiple reaction mechanisms happening simultaneously. Then $\sim_M$ is not a strict equivalent relation. However, it is always sufficient to conceptually regard the complex reaction as the combination of multiple elementary reactions where each only includes one mechanism.

[5]The version of ReactionOOD derived in this paper is v0.1.

& S$_N$2 [von Rudorff et al., 2020], RDB7 [Spiekermann et al., 2022], and Cycloaddition [Stuyver et al., 2023] has been processed by Heid et al. [2023]. However, the databases provided by Heid et al. [2023] only contain the reaction information (with atoms mapped to SMILES strings[6]) and the target properties. Thus, they can be only used for S-OOD but not C-OOD and M-OOD. We note that the E2 & S$_N$2 dataset contains reactions with molecules whose scaffold cannot be properly defined, which prevents the scaffold from being an applicable domain index for this dataset.

The well-established and open-sourced RMG kinetic databases [Johnson et al., 2022] are curated for further discussion about the C-OOD and M-OOD[7]. There are two major parts in RMG kinetic databases, i.e., REACTION FAMILY and REACTION LIBRARY. The REACTION FAMILY database contains 74 reaction mechanism families so that it can be used for the M-OOD level. The REACTION LIBRARY database contains 3,956 non-Arrhenius reactions and 3,444 pressure-dependent reactions out of 32,645 reactions in total, providing a data source for the investigation in the C-OOD level.

We use internal tools provided in the RMG databases to fetch the SMILES strings of reactants and products respectively, and integrate them to form the SMILES representation of reactions. Then, the atom mapping is extracted by RXNMapper[8] [Schwaller et al., 2021a]. For REACTION FAMILY database, we only choose the reactions with Arrhenius kinetics and set the target as the activation energy with standard units. This forms the "RMG Family" dataset at the M-OOD level in Table 2. For REACTION LIBRARY database, we further split them into two categories: reactions that depend solely on temperatures and reactions that depend on both temperature and pressure. We also filter out reactions whose valid temperature range does not contain 300K to avoid the effect of singular reactions. Then the "RMG Lib. T" and "RMG Lib. TP" datasets at C-OOD level in Table 2 are obtained by enumerating temperature in $[300K, 1, 300K, 2, 300K]$ and pressure in $[10^7, 2 \times 10^7, \ldots, 10^8]$. Rate constants at specific conditions are obtained by tools in RMG [Johnson et al., 2022] and conditions that are out of valid ranges are discarded.

For each dataset in Table 2, we follow the scheme of Gui et al. [2022] and Ji et al. [2022] to create In-Distribution (ID) and OOD splits for a comprehensive examination of the generalization abilities of existing approaches.

## 5.2 Methods

**Feature engineering.** To handle multi-molecule inputs of the kinetic property prediction task, we follow the established practice of condensed graph representation [Heid and Green, 2021] and use the graph feature extractor in a recent pipeline[9] [Heid et al., 2023]. For C-OOD settings, the temperature and pressure value is concatenated into the feature vector.

**Backbone GNN for kinetic property prediction.** All methods use the identical backbone GIN [Xu et al., 2019] with the virtual node trick. Noted that the backbone network in this paper is different from the one used in Heid et al. [2023]. However, our evaluation shows that the GIN backbone reaches similar MAE performances as [Heid et al., 2023] in the random split setting. Therefore, the vanilla Empirical Risk Minimization (ERM) [Vapnik, 1991] training is the baseline non-OOD method for comparing to other OOD methods

**OOD methods.** In addition to the ERM, we consider OOD methods developed for both Euclidean and graph data: (1) **Euclidean OOD methods** include IRM [Arjovsky et al., 2019], VREx [Krueger et al., 2021], GroupDro [Sagawa* et al., 2020], DANN [Ganin et al., 2016], Coral [Sun and Saenko, 2016], and (2) **graph OOD methods** include CIGA [Chen et al., 2022], GSAT [Miao et al., 2022] and DIR [Wu et al., 2022b]. We follow the same evaluation protocol and hyperparameter settings as in Gui et al. [2022]. Though the hyperparameters are tuned for single graph problems, they are still applicable to kinetic property prediction because the CGR feature extraction conceptually reformulates the graphs for reactants and products as one graph. We report the ID and OOD RMSE results of models selected according to the best ID and OOD validation performance. Each score is averaged by 10 runs with different random seeds.

---

[6]The Simplified Molecular-Input Line-Entry System (SMILES) is a specification in the form of a line notation for describing the structure of chemical species using short ASCII strings.

[7]https://github.com/ReactionMechanismGenerator/RMG-database

[8]https://github.com/rxn4chemistry/rxnmapper

[9]https://github.com/chemprop/chemprop

Table 3: OOD generalization performance on Cycloaddition and RDB7 dataset in two S-OOD shifts.

| | CYCLOADDITION - SCAFFOLD (S-OOD) | | | | CYCLOADDITION - TOTAL ATOM NUMBER (S-OOD) | | | |
|---|---|---|---|---|---|---|---|---|
| | COVARIATE | | CONCEPT | | COVARIATE | | CONCEPT | |
| METHODS | ID | OOD | ID | OOD | ID | OOD | ID | OOD |
| ERM | $4.78_{\pm0.07}$ | $5.80_{\pm0.40}$ | $5.04_{\pm0.06}$ | $5.57_{\pm0.08}$ | $4.12_{\pm0.07}$ | $5.23_{\pm0.29}$ | $4.30_{\pm0.11}$ | $6.00_{\pm0.13}$ |
| IRM | $13.73_{\pm0.59}$ | $16.60_{\pm1.05}$ | $17.09_{\pm0.26}$ | $18.43_{\pm0.44}$ | $17.64_{\pm0.16}$ | $17.06_{\pm0.30}$ | $23.04_{\pm0.22}$ | $22.46_{\pm0.22}$ |
| VREX | $5.50_{\pm0.06}$ | $6.41_{\pm0.86}$ | $5.17_{\pm0.05}$ | $5.76_{\pm0.06}$ | $4.84_{\pm0.07}$ | $\mathbf{4.99}_{\pm0.16}$ | $5.40_{\pm0.14}$ | $6.65_{\pm0.19}$ |
| GROUPDRO | $4.88_{\pm0.05}$ | $\mathbf{5.39}_{\pm0.55}$ | $\mathbf{5.02}_{\pm0.07}$ | $5.58_{\pm0.06}$ | $4.24_{\pm0.07}$ | $5.31_{\pm0.35}$ | $4.19_{\pm0.07}$ | $6.04_{\pm0.09}$ |
| DANN | $4.76_{\pm0.07}$ | $5.93_{\pm0.32}$ | $5.03_{\pm0.11}$ | $5.55_{\pm0.13}$ | $\mathbf{4.11}_{\pm0.10}$ | $5.17_{\pm0.23}$ | $4.22_{\pm0.07}$ | $\mathbf{5.94}_{\pm0.10}$ |
| CORAL | $\mathbf{4.75}_{\pm0.07}$ | $6.03_{\pm0.53}$ | $5.03_{\pm0.09}$ | $\mathbf{5.53}_{\pm0.11}$ | $4.17_{\pm0.07}$ | $5.25_{\pm0.20}$ | $4.26_{\pm0.11}$ | $6.02_{\pm0.07}$ |
| CIGA | $5.25_{\pm0.39}$ | $5.49_{\pm0.71}$ | $5.15_{\pm0.36}$ | $5.90_{\pm0.33}$ | $4.52_{\pm0.30}$ | $5.12_{\pm0.28}$ | $5.08_{\pm0.45}$ | $6.67_{\pm0.47}$ |
| DIR | $5.06_{\pm0.36}$ | $5.75_{\pm0.85}$ | $6.25_{\pm0.71}$ | $7.18_{\pm1.06}$ | $5.12_{\pm0.33}$ | $5.55_{\pm0.38}$ | $5.13_{\pm0.48}$ | $6.67_{\pm0.42}$ |
| GSAT | $4.81_{\pm0.07}$ | $6.02_{\pm0.17}$ | $5.04_{\pm0.08}$ | $5.59_{\pm0.09}$ | $4.17_{\pm0.13}$ | $5.90_{\pm0.13}$ | $\mathbf{4.10}_{\pm0.08}$ | $5.97_{\pm0.09}$ |

| | RDB7 - SCAFFOLD (S-OOD) | | | | RDB7 - TOTAL ATOM NUMBER (S-OOD) | | | |
|---|---|---|---|---|---|---|---|---|
| | COVARIATE | | CONCEPT | | COVARIATE | | CONCEPT | |
| METHODS | ID | OOD | ID | OOD | ID | OOD | ID | OOD |
| ERM | $9.49_{\pm0.09}$ | $22.90_{\pm1.08}$ | $10.08_{\pm0.10}$ | $\mathbf{13.67}_{\pm0.10}$ | $\mathbf{0.31}_{\pm0.00}$ | $\mathbf{0.32}_{\pm0.02}$ | $0.33_{\pm0.01}$ | $0.47_{\pm0.01}$ |
| IRM | $58.59_{\pm0.60}$ | $71.15_{\pm6.74}$ | $65.27_{\pm0.24}$ | $62.66_{\pm0.41}$ | $0.31_{\pm0.00}$ | $0.32_{\pm0.02}$ | $0.93_{\pm0.01}$ | $1.02_{\pm0.02}$ |
| VREX | $13.06_{\pm0.17}$ | $22.74_{\pm0.89}$ | $11.88_{\pm0.15}$ | $15.52_{\pm0.18}$ | $0.31_{\pm0.00}$ | $0.33_{\pm0.01}$ | $0.50_{\pm0.01}$ | $0.65_{\pm0.02}$ |
| GROUPDRO | $10.31_{\pm0.09}$ | $22.54_{\pm1.16}$ | $10.08_{\pm0.07}$ | $13.77_{\pm0.11}$ | $0.31_{\pm0.00}$ | $0.32_{\pm0.02}$ | $0.33_{\pm0.00}$ | $0.47_{\pm0.01}$ |
| DANN | $9.44_{\pm0.10}$ | $\mathbf{22.71}_{\pm0.78}$ | $10.00_{\pm0.15}$ | $13.88_{\pm0.53}$ | $0.34_{\pm0.01}$ | $0.47_{\pm0.01}$ | $0.33_{\pm0.00}$ | $0.46_{\pm0.01}$ |
| CORAL | $9.40_{\pm0.12}$ | $22.97_{\pm0.91}$ | $10.01_{\pm0.12}$ | $13.72_{\pm0.12}$ | $0.31_{\pm0.00}$ | $0.32_{\pm0.01}$ | $0.33_{\pm0.00}$ | $0.47_{\pm0.01}$ |
| CIGA | $10.48_{\pm0.86}$ | $25.05_{\pm1.74}$ | $10.99_{\pm1.09}$ | $14.23_{\pm0.86}$ | $0.32_{\pm0.01}$ | $0.39_{\pm0.04}$ | $0.36_{\pm0.01}$ | $0.49_{\pm0.03}$ |
| DIR | $10.27_{\pm0.82}$ | $24.68_{\pm2.42}$ | $13.37_{\pm1.71}$ | $16.82_{\pm1.48}$ | $0.32_{\pm0.01}$ | $0.28_{\pm0.03}$ | $0.35_{\pm0.01}$ | $0.47_{\pm0.02}$ |
| GSAT | $\mathbf{9.27}_{\pm0.11}$ | $22.72_{\pm0.30}$ | $\mathbf{9.88}_{\pm0.19}$ | $13.79_{\pm0.75}$ | $0.32_{\pm0.00}$ | $0.38_{\pm0.01}$ | $\mathbf{0.31}_{\pm0.00}$ | $\mathbf{0.32}_{\pm0.01}$ |

Table 4: OOD generalization performance on E2 & $S_N2$ and RMG with S-OOD and M-OOD shifts.

| | E2 & $S_N2$ - SIZE (S-OOD) | | | | RMG FAMILY - MECHANISM (M-OOD) | | | |
|---|---|---|---|---|---|---|---|---|
| | COVARIATE | | CONCEPT | | COVARIATE | | CONCEPT | |
| METHODS | ID | OOD | ID | OOD | ID | OOD | ID | OOD |
| ERM | $3.37_{\pm0.05}$ | $4.88_{\pm0.14}$ | $4.40_{\pm0.14}$ | $\mathbf{4.21}_{\pm0.06}$ | $29.13_{\pm0.39}$ | $118.44_{\pm6.16}$ | $32.30_{\pm0.68}$ | $\mathbf{53.25}_{\pm0.60}$ |
| IRM | $19.5_{\pm0.53}$ | $20.6_{\pm1.40}$ | $27.9_{\pm0.04}$ | $20.7_{\pm0.05}$ | $101.60_{\pm0.31}$ | $137.54_{\pm0.63}$ | $93.38_{\pm0.06}$ | $116.44_{\pm0.60}$ |
| VREX | $3.65_{\pm0.06}$ | $5.11_{\pm0.20}$ | $20.2_{\pm1.11}$ | $18.3_{\pm1.46}$ | $36.77_{\pm0.62}$ | $115.78_{\pm4.64}$ | $61.57_{\pm1.38}$ | $82.38_{\pm1.69}$ |
| GROUPDRO | $3.41_{\pm0.03}$ | $4.90_{\pm0.19}$ | $\mathbf{4.26}_{\pm0.08}$ | $4.21_{\pm0.06}$ | $\mathbf{28.11}_{\pm0.40}$ | $\mathbf{114.56}_{\pm2.77}$ | $\mathbf{31.90}_{\pm0.65}$ | $53.47_{\pm0.68}$ |
| DANN | $3.38_{\pm0.03}$ | $4.90_{\pm0.16}$ | $4.30_{\pm0.07}$ | $4.22_{\pm0.05}$ | $29.05_{\pm0.39}$ | $116.41_{\pm2.78}$ | $32.03_{\pm0.39}$ | $52.91_{\pm0.61}$ |
| CORAL | $\mathbf{3.36}_{\pm0.04}$ | $\mathbf{4.85}_{\pm0.12}$ | $4.33_{\pm0.09}$ | $4.21_{\pm0.09}$ | $29.09_{\pm0.39}$ | $116.09_{\pm3.87}$ | $31.91_{\pm0.59}$ | $52.85_{\pm0.66}$ |
| CIGA | $3.69_{\pm0.34}$ | $4.94_{\pm0.36}$ | $4.59_{\pm0.47}$ | $4.39_{\pm0.31}$ | $43.83_{\pm0.87}$ | $134.81_{\pm9.19}$ | $38.82_{\pm1.70}$ | $64.93_{\pm1.57}$ |
| DIR | $3.72_{\pm0.20}$ | $5.02_{\pm0.23}$ | $4.52_{\pm0.16}$ | $4.38_{\pm0.19}$ | $44.15_{\pm0.56}$ | $131.66_{\pm11.57}$ | $39.40_{\pm1.29}$ | $65.01_{\pm2.38}$ |
| GSAT | $3.39_{\pm0.06}$ | $5.07_{\pm0.15}$ | $4.34_{\pm0.06}$ | $4.22_{\pm0.06}$ | $30.97_{\pm0.99}$ | $117.06_{\pm3.34}$ | $32.56_{\pm0.85}$ | $53.31_{\pm0.68}$ |

## 6  Results and findings

Table 5.2 and the left half of Table 5.2 present the RMSE results for S-OOD settings, including graph size shifts on E2 & $S_N2$, Cycloaddition, and RDB7 datasets and scaffold shifts on Cycloaddition, and RDB7 datasets. The right half of Table 5.2 presents the M-OOD results of the mechanism shift on the RMG Family dataset. Table 5.2 presents the C-OOD results for T and TP shifts on the RMG Lib. dataset. For Table 5.2 and Table 5.2, the kinetic property to be predicted is the reaction barrier. For Table 5.2, the kinetic property to be predicted is the rate constant. For all tables, the best performance is boldfaced. If there are ties between ERM and any OOD methods, the ERM performance is boldfaced. Then we discuss two findings and other observations.

**Finding 1: OOD kinetic property prediction is a challenging OOD problem.** At first glance, it can be found that there remain huge ID-OOD performance gaps across different levels and types of OOD shifts and datasets. Neither OOD methods developed for Euclidean data nor OOD methods developed for graph data can outperform the vanilla ERM approaches consistently and significantly. The results also align with existing observations in realistic data and distribution shifts [Gulrajani and Lopez-Paz, 2021, Ji et al., 2022, Gui et al., 2022]. The performances of IRM are quite bad possibly due to the high requirements of IRM in learning invariant features under non-linear data [Chen et al., 2022].

**Finding 2: M-OOD shift is significantly harder than the S-OOD shift.** Moreover, it can be found that the ID-OOD performance gap is largely signified under mechanism OOD shifts (M-OOD), especially compared to the widely studied S-OOD shifts. The RMSE performances of M-OOD are An order of magnitude larger than those for S-OOD. Table 5.2 directly demonstrated this absolute

Table 5: OOD generalization performance in RMG Lib. with C-OOD shifts.

| | RMG Lib. T - T (C-OOD) | | | | RMG Lib. TP - (T, P) (C-OOD) | | | |
| | COVARIATE | | CONCEPT | | COVARIATE | | CONCEPT | |
| METHODS | ID | OOD | ID | OOD | ID | OOD | ID | OOD |
|---|---|---|---|---|---|---|---|---|
| ERM | $\mathbf{3.28}_{\pm 0.01}$ | $8.40_{\pm 0.08}$ | $\mathbf{3.70}_{\pm 0.02}$ | $6.76_{\pm 0.05}$ | $2.58_{\pm 0.01}$ | $7.35_{\pm 0.24}$ | $\mathbf{2.92}_{\pm 0.02}$ | $6.34_{\pm 0.02}$ |
| IRM | $3.41_{\pm 0.02}$ | $8.45_{\pm 0.08}$ | $7.16_{\pm 0.03}$ | $9.04_{\pm 0.02}$ | $4.30_{\pm 0.13}$ | $12.50_{\pm 0.78}$ | $9.13_{\pm 0.03}$ | $15.43_{\pm 0.06}$ |
| VREx | $3.36_{\pm 0.01}$ | $8.92_{\pm 0.13}$ | $7.27_{\pm 0.04}$ | $8.51_{\pm 0.17}$ | $\mathbf{2.55}_{\pm 0.01}$ | $7.52_{\pm 0.98}$ | $9.55_{\pm 0.13}$ | $13.10_{\pm 0.73}$ |
| GROUPDRO | $3.29_{\pm 0.02}$ | $8.45_{\pm 0.10}$ | $3.76_{\pm 0.04}$ | $6.83_{\pm 0.05}$ | $2.57_{\pm 0.01}$ | $7.39_{\pm 0.24}$ | $2.95_{\pm 0.02}$ | $6.41_{\pm 0.04}$ |
| DANN | $3.46_{\pm 0.03}$ | $8.79_{\pm 0.08}$ | $3.73_{\pm 0.04}$ | $6.82_{\pm 0.03}$ | $2.59_{\pm 0.01}$ | $\mathbf{7.16}_{\pm 0.05}$ | $2.93_{\pm 0.02}$ | $6.37_{\pm 0.03}$ |
| CORAL | $3.29_{\pm 0.02}$ | $\mathbf{8.37}_{\pm 0.06}$ | $3.75_{\pm 0.04}$ | $6.77_{\pm 0.05}$ | $2.58_{\pm 0.01}$ | $7.36_{\pm 0.16}$ | $2.94_{\pm 0.03}$ | $6.39_{\pm 0.03}$ |
| CIGA | $3.71_{\pm 0.09}$ | $8.95_{\pm 0.10}$ | $4.01_{\pm 0.05}$ | $7.12_{\pm 0.18}$ | $2.56_{\pm 0.08}$ | $7.51_{\pm 0.19}$ | $2.97_{\pm 0.05}$ | $\mathbf{6.16}_{\pm 0.04}$ |
| DIR | $3.74_{\pm 0.07}$ | $9.27_{\pm 0.05}$ | $3.96_{\pm 0.06}$ | $7.07_{\pm 0.08}$ | $3.20_{\pm 0.08}$ | $8.23_{\pm 0.44}$ | $3.11_{\pm 0.07}$ | $6.57_{\pm 0.11}$ |
| GSAT | $4.01_{\pm 0.06}$ | $9.48_{\pm 0.09}$ | $3.82_{\pm 0.05}$ | $7.06_{\pm 0.11}$ | $3.03_{\pm 0.06}$ | $8.26_{\pm 0.42}$ | $3.08_{\pm 0.03}$ | $6.58_{\pm 0.07}$ |

gap by a huge contrast, where the E2 & $S_N 2$ dataset has only two reaction mechanisms but the RMG Family dataset contains 74 reaction mechanisms.

**Miscellaneous observations.** We can also observe an exception for graph size concept shifts in E2 & $S_N 2$, that OOD performances of various methods are generically better than ID performances. One conjecture is that the distribution of molecule sizes could limit the strength of the potential concept shifts. This distinct size distribution aligns with our earlier finding that scaffolds are not always identifiable for certain molecules. For concept shifts, the ID-OOD performance gaps are mainly caused by spurious correlations across different environments or domains. For images, such spurious correlations mainly lie between the background and the object in the images. For molecules, such spurious correlations are widely observed between scaffolds and the critical functional groups [Hu et al., 2020, Koh et al., 2021]. However, for small molecules that even do not have scaffolds, we suspect there is limited room for the existence of spurious correlations. Nevertheless, it remains challenging for the OOD generalization across different graph sizes, as demonstrated by the graph size covariate shifts.

## 7 Conclusion and future works

In this work, we identified various distribution shifts exist in chemical reactions and curated a new benchmark to examine the performance of multiple OOD generalization approaches. The results show that there remain huge ID-OOD performance gaps across different distribution shifts. Therefore, it calls for better graph machine learning approaches to tackle the OOD regression challenge for facilitating the chemical proper prediction.

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
