# OpenReview forum: "Towards out-of-distribution generalizable predictions of chemical kinetic properties"
_NeurIPS.cc/2023/Workshop/AI4Science — NeurIPS2023-AI4Science Poster_

### Official Review · Reviewer_XXBr · 2023-10-21
**A useful and fascinating paper**

**Rating:** 8
**Confidence:** 4

**Review:**

This paper explores three types of domain shift in chemical reaction modeling: structure, condition and mechanism shift. While these ideas are well-known by chemists, they are often not considered by ML practitioners. Therefore, I think this work offers a useful contribution to the community. Additionally, the paper is well written with sufficient benchmarks. I have a couple questions that are at a conceptual level for future work:

- Did the authors consider the relationship between condition and mechanism shift? Often the mechanism can be shifted simply by changing the conditions. A simple example of this would be an increase in temperature leading to a decomposition reaction.
- I know there is a space constraint for these papers, but it would be nice in an extended paper to have a brief description of each of  the OOD correction methods used.

---

### Meta-Review · Area_Chair_T11a · 2023-10-27

**Recommendation:** Accept (Oral)
**Confidence:** 4

**Metareview:**

&nbsp;

I'm delighted to recommend the paper be accepted as an oral! The authors introduce a new suite of datasets and benchmarks for reaction modeling under domain shift which represents an interesting and important direction for the community.

&nbsp;

I look forward to discussing the paper at the workshop!

&nbsp;